# An Exploration of Medical Education in Central and Southern China: Measuring the Professional Competence of Clinical Undergraduates

**DOI:** 10.3390/ijerph16214119

**Published:** 2019-10-25

**Authors:** Xueyan Cheng, Jing Chen

**Affiliations:** School of Medicine and Health Management, Tongji Medical College, Huazhong University of Science and Technology, Wuhan 430030, China; xycheng6972@hust.edu.cn

**Keywords:** future physician, medical education standard, professional competence, clinical undergraduate, China

## Abstract

Background: The cultivation and assessment of the professional competence of clinical undergraduates is essential to medical education. This study aimed to construct a scale to evaluate the professional competence of clinical undergraduates as well as its determinants. Methods: The competence scale was developed on the basis of four medical education standards, the literature, and expert interviews. A total of 288 undergraduates from two types of medical colleges in central and southeastern China were selected by a multistage sampling strategy. Factor analysis, correlation analysis, and internal consistency reliability were used to verify the validity and reliability of the scale. Results: A scale consisting of eight factors with 51 items was determined for factor analysis. Cronbach’α coefficients among the eight dimensions were over 0.800, with mean scores of 1.76, 1.38, 1.92, 1.54, 1.77, 1.25, 1.60, and 2.34. Clinical undergraduates with above average academic grades achieved a higher score in essential clinical knowledge (*p* < 0.05) and better professionalism was reported among females (*p* < 0.05). Conclusion: The competence scale showed excellent reliability and validity. Respondents in this study showed a moderate level of professional competence. This study could be a reference for medical educators and policy makers in order to improve medical education standards for clinical undergraduates in China and other countries with similar settings.

## 1. Introduction

With the transition from a biological medical model to a biopsychosocial model, medical education, which traditionally focuses on essential learning in science and clinical training, has been changed to emphasize multi-dimensional competence and to meet the needs of the population [1,2,3,4]. The three widely accepted international medical education standards [5,6,7]—the Global Minimum Essential Requirements in Medical Education (GMER) in 2002, International Standards for Undergraduate Education in the World Federation of Medical Education (WFME standards) in 2001, and WHO Guidelines for Quality Assurance of Basic Medical Education in the Western Pacific Region (WHO Standards) in 2012 [8,9,10]—present this multi-dimensional perspective. In 2008, based on GMER, WFME standards, and WHO Standards, the Chinese standard Undergraduate Medical Education Standard—Clinical Medicine (trail) was implemented [11]. In 2016, the formal version was promulgated, including objectives relating to ideological, moral, and professional quality, professional knowledge, and professional skills for clinical undergraduates (Table 1) [12]. 

Furthermore, there is literature to enrich the definition and framework of professional competence from this decade. David displayed three professionalism frameworks in medical education, from virtue-based professionalism to behavior-based professionalism to professional identity formation. The last framework is viewed as an adaptive, developmental process to socialize learners into thinking, feeling, and acting like a physician at an individual as well as a collective level [13]. It is less often mentioned that professional competence is multidimensional and can be cultivated to the range of judgement and skills physicians need to be exhibit in practice [14]. Emanuel proposed that, in response to the trends of medical care, medical education should emphasize training in psychology, behavioral economics, leadership and team management, process improvement, etc. [15]. Klemenc-Ketis believed that medical professional competence should address empathy and humanism, professional relationships and development, and responsibility [16]. In competency-based medical education proposed by Powell, a health professional should be able to integrate knowledge, skills, values, and attitude [17]. However, some frameworks were theoretical frameworks and although some frameworks were used to assess the competence of students, the effectiveness of the instruments used was not verified, and it is unclear whether the outcomes of empirical studies fit the theoretical frameworks. 

Additionally, many scholars have assessed a certain dimension of the professional competence of students, such as the professionalism of medical students [18,19,20], communication skills [21,22], and medical ethics [23,24], instead of comprehensive competence. Furthermore, although some scholars have assessed the comprehensive competence of medical professionals such as physicians [25,26] and nurses [27], few studies focused on the comprehensive competence of medical undergraduates.

This study aimed to (1) construct a professional competence instrument for clinical undergraduates in China; (2) measure their professional competence; (3) explore the determinants that are associated with their competence; (4) make some suggestions for medical educators and policy makers to improve the professional competence of clinical undergraduates in China.

## 2. Methods and Materials

### 2.1. Participants and Settings

A multistage sampling strategy was used in this study. First, we selected two medical colleges purposively. College A and B are two typical medical colleges in China. College A is a crucial university, located in Wuhan, Hubei Province, in central China, and B is a non-crucial university in the city of Dongguan, Guangdong Province, in southern China. A crucial university in China refers to a university that with high education level and is support as an essential program by government. Second, a cluster sampling was used to choose clinical undergraduates. We selected 2–3 classes in each college to finish the questionnaire with the assistance of the lecturer. The investigators provided and collected the questionnaires. Also, they would explain any questions when students were confused about the questionnaire. Only five-year medical undergraduates in their last year were involved in this study. A total of 305 questionnaires were distributed and based on the validity of the questionnaires, 288 of them were included in this study. Respondents completed the questionnaire anonymously, and their competence was independent of the evaluation made by others.

In China, after passing a National College Entrance Examination administered by the Ministry of Education, students from high school can enroll as medical undergraduates. Usually, there are two kinds of medical education programs, the five-year and eight-year programs. Applicants to the eight-year program should have a higher score than those to five-year. These programs have different goals. Five-year medical students receive a bachelor’s degree when they graduate, and eight-year students receive a medical doctor’s degree. In general, it takes another three years of residency training program for five-year medical students to receive a master’s degree. Besides, they could choose to spend one year of residency training after graduation to join the Chinese Medical License Examination [28]. Thus, physicians from a five-year program account for a considerable proportion in general in China.

The respondents were investigated two months before graduation to make sure their answers to the instrument were reflect of their actual professional competence. A total of 288 medical undergraduates who majored in clinical medicine participated in this study—of which, 95 were from College A (33.0%) and 193 (67.0%) were from College B. In total, 159 (55.2%) were female and 129 (44.8%) were male. The instrument was a 4-point Likert scale (not at all/a little/most/entirely). The study protocol did not require ethical approval. The students’ information was anonymized and de-identified before the analysis.

### 2.2. Medical Competence Instrument

Based on these four standards in Table 1, we drafted a five-dimension professional competence scale, namely, dimensions of essential medical knowledge, public health and social sciences, clinical professional skills, critical thinking/adaptation, and professionalism. 

However, the professional competence of medical students is enriched with the development of the medical model and disease spectrum. We added other items according to former studies, also we modified the instrument combined with the syllabus of both college A and B and previous studies (Appendix A
Table A1). Table 2 shows the number of items from different resources in each dimension. Finally, a medical professional competence scale with 52 items in 5 dimensions was developed theoretically (Appendix A
Table A2).

To improve the reliabilty and validity of the intrument, we asked experts’ suggestion to make sure the instrument could cover the most of clinical undergraduates’ professional competence. And finally a self-assessment instrument was made to evaluate the professional competence of students [13,29].

## 3. Results

### 3.1. Factor Analysis for the Medical Professional Competence Scale

Factor analysis showed that the Kaiser-Meyer-Olkin (KMO) value of the scale is 0.939, and *p* < 0.01 in the spherical test, indicating that this scale was suitable for factor analysis. There were eight factors with a value over 1.00. One item was excluded because its factor loading was less than 0.4. Table 3 shows the status of the factor loads and variance contribution. Thus, the scale was shown to have eight dimensions with 51 items.

From Table 3, A to E in the first column showed the original construction of the instrument. The results of the factor analysis showed that the items in the dimensions of essential medical knowledge, public health and social science knowledge, critical thinking/adaptation, and professionalism were entirely consistent with the theoretical assumption. However, items in the dimension of clinical professional skill, which had been in one dimension, were divided into four dimensions. We renamed them as essential clinical skills (to understand medical history, to write medical cases correctly and other primary clinical skills), advanced clinical skills (more professional medical skills, such as the ability to adequately diagnose the patient, first aid and associated skills), communication skills (the ability to communicate well with the patient, relatives, colleagues, community and society) and advanced study skills (such as the ability to research, information retrieval skills and medical English).

### 3.2. Correlation Analysis and Internal Consistency Reliability of the Scale

In Table 4, among all the correlation coefficients of the eight dimensions, three of them were over 0.741, seven were between 0.684 and 0.611, eleven were between 0.435 and 0.596, and the rest were less than 0.362. Notably, professionalism (Factor 8) is weakly correlated with the other dimensions (0.161–0.371). Moreover, the internal consistency coefficients of each dimension were all over 0.800.

### 3.3. Medical Professional Competence of Clinical Undergraduates and Its Determinants

As Table 4, the mean scores of the dimensions of essential medical knowledge, essential clinical skills, communication skills, and professionalism was 1.76, 1.92, and 1.77, and are all higher than the total mean score. Excluding the mean score of professionalism, which was over 2.00, the mean score of all the other dimensions was between 1.25 and 1.60. Dimensions of public health and social science and advanced study skill had mean scores of 1.38 and 1.25.

Table 5 indicates that clinical undergraduates with self-rated above average academic grades had higher professional competence scores on the dimension of essential medical knowledge than those with a score that is below average (*t* = 2.406, *p* = 0.017). Respondents in College A reported better critical thinking/adaptation (*t* = 2.611, *p* = 0.010). The scores for the professionalism of female clinical undergraduate were higher than for males (*t* = 3.147, *p* = 0.002), and undergraduates in College A seemed to have greater professionalism than those in College B (*t* = 2.696, *p* = 0.007). However, there is no significant difference between the categories of each determinant in the other dimensions of the professional competence scale (*p* > 0.05).

## 4. Discussion

This study developed an instrument to evaluate the professional competence of clinical undergraduates in two medical colleges in central and southern China. The results of the factor analysis, correlation analysis, and internal consistency analysis showed instruments with excellent reliability and validity. Moreover, we also evaluated the professional competence of the respondents and analyzed the associated factors and their competence. The findings of this study provide some insights into the competence of medical students for medical educators and policy makers.

First, the professional competence scale in this study showed excellent reliability and validity. As for the eight dimensions of the scale, four of them were consistent with the theoretical assumption. The dimensions of clinical professional skills were considered as an independent dimension theoretically in this study, as well as in the medical education standards (GMER, WFME standards, and WHO standards), but factor analysis indicated that it could be divided into four independent dimensions instead of one. It could be seen in the six core competences for physicians proposed by ACGME, and dimensions such as communication skills, practice-based learning and improvement, which were considered in the dimension of clinical professional skills in this study, were also constructed as independent parts of professional competence [30]. Therefore, medical education educators should cultivate the multi-dimensional competence of students, and policy makers should improve medical standards continuously in order to meet new challenges in the field of health and medicine. 

Second, the correlation analysis suggested that professionalism is weakly related to other dimensions. In general, staff with a high level of professional knowledge and skills but weak professionalism could result in severe consequences in the field of medicine and health, as well as in other professions, such as lawyers [31] and teachers [32]. In fact, professionalism is considered an important facet when recruiting medical students in some countries. In America, medical-related social and community activities in the undergraduate period are considered as a reference when the medical colleges recruit students [33]. Since professionalism is relatively independent to other dimensions, it should be considered not only in the undergraduate period but from admission [34,35], as well as in the education period [36], internship period [37,38], and at the workplace [39]. 

Third, according to the results in Table 5, undergraduates with above average academic grades reported better essential professional knowledge scores, indicating that academic grade could only be a reflection of their professional knowledge, but it could not reflect other facets of professional competence. A scholar even suggested less preclinical training in basic sciences in response to the trend of the increase in medical information and patient data [15]. Education for clinical undergraduates should be transformed from just emphasis on knowledge and professional skill to the cultivation of comprehensive competence. Moreover, students in College A achieved higher scores in the dimensions of critical thinking/adaptation and professionalism than those in College B, which might be due to the college context, education resources, or the learning climate [40]. As for gender differences, we found that higher scores in the dimension of professionalism were reported for females than for males, which was consistent with previous studies. Investigators revealed that women possessed more compassion in their work than men [41], and female doctors were more sympathetic than male doctors [42]. Females should be treated as equally to males when college and medical institutions recruit. Thus, improving the comprehensive professional competence of medical students could be a meaningful means to improving the effectiveness of the health care system [43].

There were some limitations in this study. First, although College A and B were purposively selected, they did not represent all the medical colleges in China, and thus further studies should be conducted in more colleges. Second, a self-rated instrument was used to evaluate the medical professional competence of undergraduates, and it might not be consistent with their objective competence entirely. Finally, several potential factors, such as the college atmosphere and the education process were not addressed. 

## 5. Conclusions

A medical professional competence instrument was developed in this study to the evaluate professional competence of clinical undergraduates in central and southern China. According to the results of the factor analysis, the correlation analysis, and the internal consistency reliability, this instrument has excellent reliability and validity. The professional competence scale consisted of eight dimensions with a total of 51 items. These dimensions were essential medical knowledge, public health or social science, essential clinical skills, advanced clinical skills, communication skills, advanced study skills, critical thinking and adaptation, and professionalism. Respondents in this study showed a moderate level of professional competence. Their public health/social sciences knowledge, advanced clinical skills, and critical thinking/adaptation still need to be improved. Undergraduates with an above average academic grade achieved higher scores in the dimension of essential medical knowledge than those with a below average academic grade. Undergraduates in College A achieved a higher score in the dimension of critical thinking/adaptation and professionalism than those in College B. Moreover, better professionalism was reported among females. This study could be a reference for medical educators and policy makers in order to improve medical education standards for clinical undergraduates in China and other countries with similar settings.

## Figures and Tables

**Table 1 ijerph-16-04119-t001:** Dimensions of four medical education standards.

	Global Minimum Essential Requirements in Medical Education (GMER)	International Standards for Undergraduate Education in the World Federation of Medical Education (WFME Standards)	WHO Standards	Chinese Standards
**Dimension**	Professional values, behavior and ethics	Essential biomedical sciences	General objectives	Objectives relating to ideological and moral and professional quality
	Scientific foundation of medicine	Behavior, social sciences and medical ethics	Objectives relating to knowledge	Objectives relating to knowledge
	Communication skills	Clinical sciences and skills	Objectives relating to skills	Objectives relating to skills
	Clinical skills		Objectives relating to professionalism	
	Population health and health systems;			
	Management of information;			
	Critical thinking and research			

**Table 2 ijerph-16-04119-t002:** Number of items from different resources in each dimension.

Dimensions of the Scale in This Study	Total Number in Each Dimension	GMER	WHO Standards	WFME Standards	Chinese Standards	Others
Essential medical knowledge	4	4	4	4	4	0
Public health and social science knowledge	12	3	2	3	11	1
Clinical professional skills	17	11	11	5	9	2
Critical thinking/adaptation	7	3	1	1	3	2
professionalism	12	0	3	0	6	3

**Table 3 ijerph-16-04119-t003:** Factor analysis of medical professional competence scale.

Items	Factors							
	F1	F2	F3	F4	F5	F6	F7	F8
**A1**	0.365	0.163	0.241	0.199	−0.005	0.121	0.435	−0.010
**A2**	0.280	0.208	0.190	−0.006	0.321	0.135	0.666	0.049
**A3**	0.217	0.160	0.204	0.008	0.377	0.095	0.690	−0.014
**A4**	0.398	0.125	0.109	0.107	0.093	0.192	0.600	0.115
**B5**	0.554	0.146	0.107	0.365	0.002	0.204	0.395	−0.059
**B6**	0.612	0.063	0.189	0.327	0.084	0.223	0.301	−0.012
**B7**	0.699	−0.018	0.163	0.192	0.016	0.160	0.274	0.136
**B8**	0.700	−0.062	0.090	0.053	0.113	0.097	0.118	0.173
**B9**	0.738	0.069	0.106	0.089	0.169	0.094	0.062	0.080
**B10**	0.642	0.247	0.110	0.186	0.199	0.014	0.101	0.061
**B11**	0.764	0.009	0.117	0.155	0.113	0.094	0.039	0.110
**B12**	0.738	0.074	0.111	0.096	0.083	0.103	0.147	0.102
**B13**	0.802	0.041	0.140	0.122	−0.011	0.113	0.084	0.075
**B14**	0.805	−0.004	0.168	0.099	0.073	0.143	−0.012	0.117
**B15**	0.717	0.119	0.289	−0.049	0.057	0.136	0.085	0.050
**B16**	0.667	0.073	0.274	−0.016	0.185	0.100	0.177	0.112
**C17**	0.196	0.205	0.275	0.310	0.664	0.086	0.185	−0.012
**C18**	0.182	0.278	0.154	0.165	0.769	0.075	0.092	−0.012
**C19**	0.191	0.121	0.130	0.150	0.752	0.183	0.201	0.079
**C20**	0.312	0.096	0.173	0.183	0.222	0.554	0.268	0.140
**C22**	0.298	0.083	0.192	0.273	0.354	0.401	0.219	0.234
**C23**	0.355	0.034	0.229	0.383	0.122	0.541	0.125	0.104
**C24**	0.324	0.086	0.121	0.215	0.244	0.552	0.186	0.143
**C25**	0.286	0.105	0.259	0.305	0.162	0.551	0.207	−0.019
**C30**	0.304	−0.009	0.264	0.151	−0.043	0.536	−0.026	0.292
**C26**	0.200	0.153	0.223	0.579	0.302	0.343	0.060	0.007
**C27**	0.216	0.264	0.220	0.585	0.331	0.092	0.014	−0.030
**C28**	0.257	0.100	0.210	0.687	0.090	0.196	0.113	0.170
**C29**	0.193	0.155	0.205	0.659	0.182	0.235	0.041	0.142
**C31**	0.478	0.017	0.238	0.077	−0.083	0.412	0.042	0.510
**C32**	0.401	0.056	0.258	0.058	0.071	0.277	0.100	0.621
**C33**	0.336	0.056	0.216	0.121	0.071	0.107	0.035	0.730
**D34**	0.431	0.012	0.539	0.151	−0.002	0.203	−0.092	0.407
**D35**	0.337	0.025	0.683	0.161	0.075	0.146	0.043	0.223
**D36**	0.291	0.060	0.755	0.200	0.066	0.139	0.105	0.145
**D37**	0.289	0.174	0.796	0.121	0.158	0.134	0.094	0.002
**D38**	0.260	0.172	0.735	0.161	0.105	0.141	0.171	0.072
**D39**	0.153	0.286	0.579	0.183	0.311	0.052	0.119	0.098
**D40**	0.091	0.277	0.601	0.035	0.147	0.111	0.230	0.049
**E41**	0.106	0.577	0.068	0.193	−0.141	−0.025	0.353	0.058
**E42**	0.130	0.759	−0.031	0.263	−0.021	−0.043	0.188	0.253
**E43**	0.148	0.771	0.002	0.281	0.088	−0.015	0.152	0.203
**E44**	0.095	0.812	0.048	0.197	0.064	−0.155	0.093	0.211
**E45**	0.046	0.815	0.111	0.169	0.032	−0.135	0.131	0.147
**E46**	0.015	0.817	0.127	0.151	0.062	−0.112	0.053	0.048
**E47**	0.062	0.784	0.136	0.042	0.110	0.006	−0.019	−0.036
**E48**	−0.026	0.821	0.098	−0.098	0.067	0.114	0.035	−0.222
**E49**	0.002	0.837	0.125	−0.010	0.114	0.161	0.039	−0.146
**E50**	−0.019	0.864	0.107	−0.022	0.118	0.216	−0.001	−0.124
**E51**	−0.009	0.857	0.057	−0.103	0.146	0.158	−0.017	−0.089
**E52**	0.114	0.805	0.098	−0.068	0.147	0.198	0.008	0.068

Note: The variance contribution of the first eight factors reached 68.086%; shaded tables in the same column were considered as in the same factor. Theoretical assumption: A—essential medical knowledge; B—public health and social sciences; C—clinical professional skills; D—critical thinking/adaptation; E—professionalism. Factor analysis results: F1—essential medical knowledge; F2—public health and social science; F3—essential clinical skill; F4—advanced clinical skill; F5—communication skill; F6—advanced study skill; F7—critical thinking/adaptation; F8—professionalism. KMO: Kaiser-Meyer-Olkin.

**Table 4 ijerph-16-04119-t004:** Correlation analysis and the internal consistency reliability of the scale.

Dimensions	F1	F2	F3	F4	F5	F6	F7	F8	Total	Cronbach’s α	X ± S
**F1**	1									0.800	1.76 ± 0.54
**F2**	0.636 *	1								0.939	1.38 ± 0.61
**F3**	0.534 *	0.449 *	1							0.852	1.92 ± 0.51
**F4**	0.572 *	0.684 *	0.542 *	1						0.852	1.54 ± 0.64
**F5**	0.498 *	0.533 *	0.572 *	0.665 *	1					0.842	1.77 ± 0.59
**F6**	0.435 *	0.643 *	0.272 *	0.630 *	0.466 *	1				0.833	1.25 ± 0.70
**F7**	0.538 *	0.611 *	0.501 *	0.652 *	0.581 *	0.595 *	1			0.899	1.60 ± 0.65
**F8**	0.362 *	0.214 *	0.373 *	0.237 *	0.345 *	0.161 *	0.338 *	1		0.952	2.34 ± 0.57
**Total**	0.741 *	0.827 *	0.663 *	0.809 *	0.744 *	0.681 *	0.809 *	0.596 *	1	0.962	1.70 ± 0.60

Note: * *p* < 0.05; F1—essential medical knowledge; F2—public health and social science; F3—essential clinical skill; F4—advanced clinical skill; F5—communication skill; F6—advanced study skill; F7—critical thinking/adaptation; F8—professionalism. X mean score; S Standard deviation.

**Table 5 ijerph-16-04119-t005:** Determinants are associated with the professional competence of respondents.

Dimensions	Determinants		*n*	Scores (X ± S)	*t*	*p*
**Essential medical knowledge**	Academic grade	Below average	153	6.82 ± 1.717	2.406	0.017
		Above average	135	7.30 ± 1.658		
**Critical thinking/adaptation**	College	College A	95	4.147 ± 1.930	2.611	0.010
		College B	193	3.568 ± 1.689		
**Professionalism**	Gender	Male	129	27.02 ± 5.782	3.147	0.002
		Female	159	29.05 ± 5.140		
	College	College A	95	29.38 ± 5.728	2.696	0.007
		College B	193	27.53 ± 5.325		

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
