# Peer review of "An Exploration of Medical Education in Central and Southern China: Measuring the Professional Competence of Clinical Undergraduates"

_ijerph, 2019, doi:10.3390/ijerph16214119_

Round 1

Reviewer 1 Report

The paper presented for the review concerns the construction of a scale for clinical undergraduates to evaluate their professional competence and its associated determinants. This paper is well-written and easy to follow. However, there are some remarks that have to be addressed:

All the text needs a minor language revision in order to improve some English nuances. I suggest to stress, at least briefly, how the learning climate may influence significantly the quality of training among medical students/residents and how this could affect their professional competence in the future (refer to: PMID: 31437620; PMID: 27260008). I also suggest mentioning the effectiveness of virtual simulators in increasing the diagnostic and surgical skills of medical students/residents. Some interesting studies about the topic are the following: PMID: 31169414; PMID: 25463761; PMID: 24056136.

Author Response

Dear reviewer,

Thank you very much for your useful recommendations on my manuscript. I read them carefully and here is my response:

Point 1: Learning climate

Response 1: I read the literature you provided carefully, and I notice that the learning climate may play an important role in medical education in some countries. Enrico indicated that better formal teaching and specialty tutors ensure better education, and it would be more likely to improve students’ professional competence. Medical education and students’ competence could not be improved for lacking of good learning climate in many schools, regions and countries. So I mentioned it in the section of discussion.

Point 2: About the effectiveness of virtual simulators in medical education

Response 2: I noticed that Salvatore’s study proved the effectiveness of virtual simulators in increasing the diagnostic and surgical skills, however it was independent from their starting level of expertise. Therefore, it might be a useful way to improve students’ competence in further education/workplace or in practice curriculum. So I did not discuss this point of view in my manuscript, but it would be helpful in my future study.

At last, I also modified the language, introduction, results and conclusion of the manuscript.

If there are still some problems, please let me know.

Thank you again for reviewing my manuscript.

Have a nice day!

Yours,

Xueyan Cheng

Reviewer 2 Report

Thank you for your submission of the manuscript to International Journal of Environmental Research and Public Health. This seems a study to evaluate the students' competence by using self-administered questionnaire.

The assessment of a medical student's professional competence should be conducted from the viewpoint of professional standard, because unqualified students for future clinical practice do not seem to be qualified to assess their own competence. So I do not understand how you can prove that students' competence is sufficient for future practice.

As for the detailed point of the methods, you have to mention how you provide and collect the questionnaire. Readers do not understand if this questionnaire is associated with assessment of each student or not. such background information is significantly important for the study of assessment.

Author Response

Dear reviewer,

Thank you very much for your useful recommendations on our manuscript. We read them carefully, and here is our response:

Point 1: The assessment of a medical student’s professional competence should be conducted from the viewpoint of a professional standard

Response 1: First, the assessment of medical students’ professional competence in this study was constructed based on three widely accepted international medical education standards (GMER; WFME standards; WHO standards) and the Chinese medical standard. Items of the scale were from these four medical education standards mostly, and considering changes in medical occupation and other facets. Based on previous studies, we pick 8 items that not in the standards but are also crucial for students’ competence to complete the instrument (Appendix 1). Second, suggestions from doctors and experts in this field were considered to modify the competence scale. We also made a focus group to make sure it could be covered with the most of professional competence for clinical undergraduates, and to make sure all the items were easy to understand for all respondents. At last, the empirical research proved the scale to be of excellent reliability and validity.

Point 2: Unqualified students for future clinical practice do not seem to be qualified to assess their own competence

Response 2: Medical professional competence of the clinic was often measured by others or by themselves (PMID: 27524927; PMID: 21838887). Considering that there are many dimensions in the scale, including clinical knowledge, and practice skills, clinical-related knowledge and practice skills. Besides, there are some other competency in the scale which would be difficult to evaluate by others. Therefore, we decided to build a self-rating scale in this study.

Point 3: How to prove that students' competence is sufficient for future practice

Response 3: First, to make the manuscript more easily to understand, I changed the title from “future physician” to “five-year clinical students, because only five-year clinical students were involved in this study.

Second, students would obtain more professional competence in further education and the workplace. For example, virtual simulator was effective in increasing the diagnostic and surgical skills of medical students, which was independent from their starting level of expertise (PMID: 31169414). Furthermore, specific medical standards were also made for postgraduates and eight-year students in China, and there would be more practice curriculum for clinical students in further education. Therefore, it is a long period to improve students’ professional competence to adapt future needs, and this study would provide some new insight for further education of five-year clinical undergraduates.

Point 4: How I provide and collect the questionnaire

Response 4: I provided and collected the questionnaires with the assistance of the lecturers. The students had enough time to complete the questionnaire during the class. We would explain all the problems when students have some uncertain questions. I introduced the process in the section of methods.

Poind 5: If this questionnaire is associated with assessment of each student or not

Response 5: Students completed the questionnaire anonymously, and the assessment on them was independent from assessment in other ways.

At last, I also modified the language, introduction, results, and conclusion of the manuscript.

If there are still some problems, please let me know.

Thank you again for reviewing my manuscript.

Have a nice day!

Yours,

Xueyan Cheng

Reviewer 3 Report

I read with great interest the Manuscript titled “An Exploration of Medical Education in Central and Southern China: Measuring Professional Competence for Future Physician” (ijerph-592142), which falls within the aim of International Journal of Environmental Research and Public Health.     

In my honest opinion, the topic is interesting enough to attract the readers’ attention. Methodology is accurate and conclusions are supported by the data analysis.

Authors should consider only the following recommendations:

Manuscript should be further revised by a native English speaker in order to correct several typos. References should be formatted according to the Journal’s guidelines.

Author Response

Dear reviewer,

Thank you very much for your useful recommendations on my manuscript. I read them carefully and here is my response:

Point 1: to revise the language of the manuscript

Response 1: I have modified the language by a native English speaker, and the typos were corrected. 

And I also modified the language, introduction, results and conclusion of the manuscript.

If there are still some problems, please let me know.

Thank you again for reviewing my manuscript.

Have a nice day!

Yours,

Xueyan Cheng

Round 2

Reviewer 2 Report

Thank you for your submission of the manuscript to the journal. There are many articles of assessment of professionalism in medical education field. Citation of such articles are not thorough. Especially key articles are not cited here. If this content is useful domestically, please switch the manuscript for such purpose. 

Author Response

Dear reviewer,

Thank you very much for your recommendations on our manuscript. We read them carefully and here is our response:

Point 1: There are many articles of assessment of professionalism in medical education field. Citation of such articles are not thorough. 

Response 1: Key words, such as “professional competence”, “professionalism assessment/evaluation” and “medical education”, were used to search articles in Pubmed. We read some high-cited articles and their references, and cite five relevant articles in the manuscript (reference 3, 4, 13, 14, 29). I also deleted two which I thought were redundant (reference 13, 14 in the former version). Besides, I changed two references to a more suitable ones (reference 22, 23).

If there are still some problems, please let me know.

Thank you again for reviewing my manuscript.

Have a nice day!

Yours,

Xueyan Cheng